# Physical Activity during Pregnancy: Comparisons between Objective Measures and Self-Reports in Relation to Blood Glucose Levels

**DOI:** 10.3390/ijerph19138064

**Published:** 2022-06-30

**Authors:** Hanqing Chen, Xuanbi Fang, Tak-Hap Wong, Sze Ngai Chan, Babatunde Akinwunmi, Wai-Kit Ming, Casper J. P. Zhang, Zilian Wang

**Affiliations:** 1Department of Obstetrics and Gynaecology, The First Affiliated Hospital, Sun Yat-sen University, Guangzhou 510060, China; chenhanq@mail.sysu.edu.cn; 2Department of Infectious Diseases and Public Health, Jockey Club College of Veterinary Medicine and Life Sciences, City University of Hong Kong, Hong Kong, China; fangxuanbi@stu2021.jnu.edu.cn (X.F.); thwong363-c@my.cityu.edu.hk (T.-H.W.); 3Department of Obstetrics and Gynaecology, The First Affiliated Hospital of Jinan University, Guangzhou 510632, China; 1155123639@link.cuhk.edu.hk; 4Center for Genomic Medicine (CGM), Massachusetts General Hospital, Harvard Medical School, Harvard University, Boston, MA 02142, USA; bakinwunmi@bwh.harvard.edu; 5Department of Obstetrics and Gynecology, Brigham and Women’s Hospital, Harvard Medical School, Harvard University, Boston, MA 02142, USA; 6School of Public Health, LKS Faculty of Medicine, The University of Hong Kong, Hong Kong, China

**Keywords:** gestational diabetes mellitus, blood sugar, fasting plasma glucose, pregnant, international physical activity questionnaire

## Abstract

Objective: to quantify pregnant women’s physical activity during pregnancy using wearable accelerometers and a self-reported scale and to examine the variation in these two physical activity measures in relation to fasting plasma glucose (FPG) levels and presence of gestational diabetes mellitus (GDM). Methods: this prospective observational study included 197 pregnant women from one of the largest regional hospitals in South China. Women with singleton pregnancy, absence of pre-existing comorbidities and pre-specified contraindications, wore an accelerometer on their waist for 7 consecutive days to objectively record their physical activity, followed by completing a past-7-day physical activity questionnaire, three times, respectively, in three trimesters. GDM was determined by 2-h 75 g oral glucose tolerance test (OGTT) in 24–28th week’s gestation and FPG was obtained in both 1st and 2nd trimesters following standard practice. Results: pregnant women engaged highest levels of various physical activity types in 2nd trimester, except accelerometer-based moderate-to-vigorous physical activity which gradually decreased in pregnancy. Pregnant women were more likely to walk in 3rd trimester. The relationship between objective total physical activity and self-reported total physical activity was non-linear. Increased trend of FPG from 1st trimester to 2nd trimester disappeared when adjusting for accelerometer-based light physical activity and attenuated when including walking. Self-reported moderate physical activity was surprisingly positively associated with GDM. Conclusions: different patterns in physical activity between objective measure and self-report in relation to gestational glucose levels were observed. Short-term increase in moderate physical activity prior to OGTT may not be necessary for reducing presence of GDM. Future glucose management for pregnant women may be targeted at lower intensity physical activity.

## 1. Introduction

Physical activity during pregnancy is considered beneficial to pregnant women for optimal pregnancy outcomes. According to the World Health Organisation’s physical activity guidelines, pregnant women without contraindication are recommended to engage in regular physical activity with at least 150 min weekly of moderate physical activity (MPA) to obtain health benefits [1]. To effectively act on the recommendation, quantifying pregnant women’s physical activity is essential.

Commonly, physical activity is measured subjectively by self-reports or objectively using wearable devices. Self-reports such as self-administered questionnaires provide perception-based physical activity estimates accounting for the variation across trimesters in pregnant women. Although the physical activity intensity can reflect personal physical capability, subjective bias is inevitable [2,3]. Wearable devices that automatically capture different intensities of physical activity undertaken by pregnant women allow for more accurate estimates [4]. The quality of estimation relies on the intensity category (e.g., cut-points) appropriate for specific populations [5]. To date, physical activity intensity classification established for pregnant women is not available and the cut-points for general adults are commonly used for pregnant women [6]. It is important to understand the relationship between physical activity measured by these two methods to inform the establishment of device-based intensity classification for pregnant women and thus potential physical activity prescription by clinicians.

Multiple pregnancy-related conditions, including blood glucose levels, can benefit from physical activity [7,8]. Higher blood glucose during pregnancy and related diagnosis of gestational diabetes mellitus (GDM) are conditions that can lead to poor pregnancy outcomes and long-term adverse consequences to both the mother and the offspring [9,10]. Previous studies mainly examined self-reported physical activity and there were mixed findings on the effect of physical activity or exercise on GDM [11]. A recent systematic review [11] identified six observational studies examining the effect of physical activity during pregnancy on GDM, and all except one [12] employed self-reported measures and only one study reported significant findings [13]. The only study employed both objective and self-reported measures but unfortunately only asked participants to report an overall intensity of regular physical activity [12], which precludes an intensity-to-intensity comparison between two measures. In those intervention studies [14], the number of exercise sessions that participants attended was measured. However, objective measures of physical activity per se were scarcely employed and the intervention compliance with targeted exercise intensity was thus uncertain due to not being accurately assessed. Only a handful of studies examined objectively measured physical activity in pregnancy [12,15,16] and one study [12] found increased armband-recorded steps in early pregnancy associated with lower occurrence of GDM.

In consideration of the above, this study aimed to quantify pregnant women’s physical activity measured by wearable accelerometers and a self-reported scale and to examine the variation in these two physical-activity measures in relation to fasting glucose levels and the risk of GDM.

## 2. Materials and Methods

### 2.1. Study Design and Participants

This prospective observational study was to quantify physical activity during pregnancy using both accelerometers and a validated questionnaire and to examine the associations of physical activity with the risk of GDM and fasting glucose levels.

Participants were recruited from the First Affiliated Hospital of Sun Yat-sen University, one of the largest regional hospitals in South China, who were attending their first antenatal check-up between the 10–14th gestational weeks. All potential participants were initially evaluated by clinicians for their eligibility for participation. Pregnant women were enrolled for participation if they were (1) at 10^+0^ to 14^+6^ weeks’ gestation, (2) aged between 18–40 years, (3) singleton pregnancy with no foetal abnormality determined by ultrasound scans. Exclusion criteria were (1) twin or multiple pregnancy, (2) pre-existing comorbidities including hypertension, type 1 or type 2 diabetes, thrombocytopenia, etc., (3) history of cervical insufficiency or shortening of uterine cervical canal in this pregnancy, (4) threatened miscarriage, placenta previa, or any abnormal uterine bleeding in this pregnancy, (5) malignant tumours and (6) any mental disorders.

Eligible participants, upon provision of their informed consent to participation, were asked to wear an accelerometer at their waist for 7 consecutive days to record their physical activity, followed by completing a questionnaire about their physical activity in the past 7 days, three times in their 1st (10–14th week), 2nd (20–24th week) and 3rd (30–34th week) trimesters. The study was undertaken during March 2018 to September 2019. The study was reviewed and approved by the Ethics Committee of the First Affiliated Hospital of Sun Yat-sen University (Ref: (2017)296).

### 2.2. Measures

#### 2.2.1. Physical Activity

Objective physical activity was assessed using an accelerometer ActiGraph GT3X+ (ActiGraph Inc., Pensacola, FL, USA). Participants were instructed to wear at their waist 24 h for 7 consecutive days, except for sleep and activities in water. The ActiGraph’s data were expressed as counts per minute (cpm), and sixty-second epoch was specified for running algorithms. Non-wear time was defined as 60 consecutive minutes of zero cpm. Those wears with at least 10 h wear time on 5 days (at least four weekdays plus one weekend) were considered as valid. Upon validation, if not valid, participants would be asked to wear for additional days (weekday and/or weekend, dependent on the number of days not valid) starting on the following day. Freedson’s adult cut-points [17] were used to determine physical activity intensity: <100 cpm (sedentary), 100–1951 cpm (light physical activity), 1952–5724 cpm (MPA), and ≥5725 cpm (vigorous physical activity). Physical activity variables of different intensities were reported in minutes per week. 

Self-reported physical activity was assessed using the Chinese version of self-administered International Physical Activity Questionnaire (IPAQ)–Long Form [18]. IPAQ has been validated in multicultural adult population and widely used among pregnant women [19]. In this study, we calculated the weekly minutes of walking, non-walking MPA, total MPA (added up by walking plus non-walking MPA), and total moderate-to-vigorous physical activity (MVPA) for comparison purpose with objective physical activity. 

#### 2.2.2. Gestational Diabetes Mellitus (GDM)

GDM was determined by 2-h oral glucose tolerance test (OGTT) with 75 g glucose performed in 24–28th week’s gestation [20]. Diagnosis of GDM was made if a pregnant woman obtained one abnormal value of three thresholds: ≥5.1 mmol/L for fasting, ≥10.0 mmol/L for 1-h, and ≥8.5 mmol/L for 2 h [21].

#### 2.2.3. Fasting Blood Glucose (FPG)

FPG levels were obtained in both 1st and 2nd trimesters in line with standard practice. The FPG value of 2nd trimester was derived from the fasting of OGTT undertaken in 24–28th week’s gestation. Pregnant women were instructed to fast overnight for at least 8 h prior to the test. 

#### 2.2.4. Other Information

Information on participants’ demographic characteristics and pregnancy- and health-related factors was also collected. The information included age (year), gestation (week), educational attainment (no formal education, primary, secondary, or higher education), occupation (full-time, part-time, or unemployed), household monthly income (high or low), primipara (yes or no), in vitro fertilisation (yes or no), miscarriage history (yes or no), and pre-pregnancy body mass index (BMI, kg/m^2^). 

### 2.3. Analytical Approaches

It is estimated that a sample of 194 would allow for detecting a small-to-moderate correlation (approximately 0.20) between objectively-measured and self-reported physical activity, with an 80% statistical power and using two-tailed probability level of 0.05 [22]. In consideration of a potential withdrawal rate of 8–10% in follow-up, we aimed to recruit a sample of 211–216 pregnant women consenting to participation. By 1st trimester, 212 women successfully provided valid data of both accelerometer and IPAQ. In 2nd and 3rd trimesters, respectively, four and eight participants failed to provide valid accelerometer data or withdrew due to other medical issues. We further excluded three participants with very extreme/unreasonable values of IPAQ (as specified in the IPAQ’s data cleaning instructions) [23]. We finally included the data of 197 pregnant women in analysis.

Descriptive statistics were computed for all variables. Frequencies and percentages were reported for categorical variables. Means and standard deviations were reported for all continuous variables; medians and interquartile ranges were additionally reported for those variables that were not normally distributed, with skewness >1 or <−1. 

Since accelerometer-based physical activity variables were right-skewed, non-negative and continuous values, generalised linear models with gamma variance and logarithmic link functions [24], being more appropriate than general linear models, were used to examine the associations of accelerometer-based physical activity outcome variables with demographics and other pregnancy- and health-related factors.

Negative binomial models were used to estimate the associations of self-reported physical activity outcome variables (modelled as count variables given the nature of IPAQ as well as their distributions) with demographics and other pregnancy- and health-related factors. Negative binomial models with robust standard errors accounting for between-individual difference were used to estimate differences in physical activity outcome variables across trimesters, and predicted margins were obtained for each trimester and shown in plots as appropriate. For those outcome variables with a large number of zero values (i.e., self-reported non-walking MPA, walking across trimesters) than expected under standard negative binomial models (determined by Akaike information criterion and Bayesian information criterion) were replaced by zero-inflated negative binomial models. Zero-inflated negative binomial models yield two sets of regression estimates: (1) Odds ratios of non-zero values to zero values of specific physical activity outcomes (i) between different groups (categorical variables, e.g., lower vs. higher education) or (ii) associated with 1-unit changes in predictors (continuous variables, e.g., BMI) and (2) the proportional differences (reported as antilogarithm of regression coefficients) in non-zero outcome values between different groups or associated with 1-unit changes in predictors. Only physical activity variables in 2nd trimester (prior to OGTT) were analysed in this study because their short-term relationships with GDM and FPG assessed in 2nd trimester were assumed more profound than those physical activity variables of other trimesters. 

To examine the potential effects of demographics and physical activity on glucose-related outcomes, logistic regressions were used to model the associations between presence of GDM and predictors (e.g., demographics and physical activity variables) while general linear models were used to model the association of FPG values (continuous and normally distributed) with predictors. 

All above-mentioned models were repeated for estimating covariate- and/or confounders-adjusted associations between variables of interest. Analyses above were conducted in Stata 16.0.

Given that progressive abdominal enlargement during gestation can affect physical activity, universal accelerometer cut-points of physical activity intensity for general adults may not be appropriate for characterising pregnant women’s objective physical activity across trimesters. We thus examined two associations, respectively, (1) between accelerometer-based MVPA and self-reported MVPA and (2) between accelerometer-based total physical activity and self-reported MVPA (i.e., total physical activity measured by IPAQ), via regressing objective physical activity on self-reported MVPA. Generalised additive mixed models are able to model associations with unknown forms in correlated data [25]). In this study, generalised additive mixed models with gamma variance and logarithmic link functions accounting for between-individual difference were thus employed with a smooth term (thin-plate spline method) used for modelling a potential non-linear association. These analyses were conducted in R (version 4.1.2, R Foundation for Statistical Computing, Vienna, Austria) using the packages ‘mgcv’ and ‘gmodels’.

## 3. Results

### 3.1. Sample Characteristics

Table 1 shows the sample characteristics. This sample included 197 pregnant women with three observation points across three trimesters [1st trimester (T1) = 12.6 (±0.90) weeks, 2nd trimester (T2) = 21.7 (±1.04) weeks and 3rd trimester (T3) = 31.0 (±0.75) weeks]. The majority were degree holders or above (90.9%), employed full-time (80.7%), primipara (70.6%), not under in-vitro-fertilisation (85.3%), and free from miscarriage history (80.7%). The sample pre-pregnancy BMI was 20.9 (±3.05) kg/m^2^ and their 1st trimester FPG value was 3.99 (±0.41) mmol/L. The fasting, 1-h and 2-h readings of OGTT were 4.20 (±0.32), 7.39 (±1.63) and 6.44 (±1.44), respectively, yielding 19 pregnant women (9.6%) diagnosed of GDM. 

### 3.2. Physical Activity during Pregnancy

Table 2 shows the details of both accelerometer-based and self-reported physical activity variables by trimester. Overall, across the three trimesters there was a trend that highest objective physical activity was observed at 2nd trimester, then 1st trimester followed by 3rd trimester, although the differences were not statistically significant except for accelerometer-based MVPA. For accelerometer-based MVPA, this type of physical activity dropped across trimesters and reached the lowest in 3rd trimester (1st trimester vs 3rd trimester, *p* = 0.009; also see Figure 1). Self-reported physical activity also saw the highest level at 2nd but 1st trimester the lowest. Similarly, the differences between trimesters in terms of duration were not statistically significant. However, pregnant women were more likely to engage any walking in their 3rd trimester than in 1st trimester (*p* = 0.027; also see Figure 2).

### 3.3. Accelerometer-Based Physical Activity versus Self-Reported Physical Activity

As the identified relationship between accelerometer-based MVPA and self-reported MVPA indicates poor agreement (Appendix A), we reported the relationship between accelerometer-based total physical activity and self-reported MVPA (i.e., total self-reported physical activity or termed as ‘total MVPA’ to differentiate accelerometer-based ‘MVPA’) in detail. The relationship between accelerometer-based total physical activity and self-reported total physical activity was non-linear (Figure 3; *F*_2.366_ = 24.84, *p* < 0.001), whereby a steeper slope was observed at lower self-reported total physical activity. By adding a reference line to indicate the absolute agreement, we found that when pregnant women reported fewer than ~1800 min of total physical activity per week, they in fact accumulated higher amount of accelerometer-based physical activity than their self-report. In contrast, those who reported higher than ~1800 min per week, they accrued fewer accelerometer-based physical activity than their self-report. The pattern held the same across trimesters. 

### 3.4. Demographic and Pregnancy-Related Correlates of Physical Activity Variables

As to accelerometer-measured physical activity (Appendix A), women who were underweight prior to pregnancy, compared to those with normal BMI, engaged in less light physical activity in 2nd trimester (e*^b^* = 0.883, 95% CI 0.803 to 0.972, *p* = 0.011). Higher-educated and full-time employed women and those without in vitro fertilisation undertook more MVPA compared with their counterparts (see Appendix A). For total physical activity, women with higher pre-pregnancy BMI were likely to engage more physical activity in 2nd trimester (1.3% more physical activity as 1 kg/m^2^ increases), contributed by underweight women undertaking less physical activity than those at normal weight (e*^b^* = 0.883, 95% CI 0.808 to 0.964, *p* = 0.006). 

For self-reported physical activity, women with higher education and higher household income reported less walking, total MPA and total MVPA in 2nd trimester than their counterparts (see Appendix A). Full-time employed women reported more walking than those not employed full-time (e*^b^* = 1.546, 95% CI 1.020 to 2.342, *p* = 0.040). Women who were underweight prior to pregnancy reported less total MPA and MVPA in 2nd trimester than those with normal weight (Appendix A). As to non-walking MPA, women with higher pre-pregnancy BMI were likely to undertake any non-walking MPA in 2nd trimester (adjusted *OR* = 1.214, 95% CI 1.062 to 1.388, *p* = 0.005), resulting from those underweight being less likely to engage any compared to those with normal weight. It is notable that full-time employed women who engaged in any non-walking MPA in 2nd trimester reported less amount than their counterparts not employed full-time (e*^b^* = 0.470, 95% CI 0.278 to 0.783, *p* = 0.004).

### 3.5. Demographic and Health-Related Correlates of GDM and FPG Level

Before examining the covariate-adjusted associations of physical activity with glucose-related outcomes, we first compared the risk of GDM and the FPG values (in both 1st and 2nd trimester) between pregnant women with different demographic characteristics (Table 3). We found that those higher-educated were less likely develop GDM compared to their lower-educated counterparts (adjusted *OR* = 0.211, 95% CI 0.047 to 0.954, *p* = 0.043). Those with higher pre-pregnancy BMI had higher FPG values at both 1st and 2nd trimesters, with 2.5% and 3.2%, respectively, increased risks of elevated FPG levels as 1 unit of BMI increased (see Table 3 for details). There was also an average 0.123 mol/L (13%) increase in FPG value in 2nd trimester compared to that in 1st trimester, adjusted for demographics and other covariates (*b* = 0.123, 95% CI 0.012 to 0.233, *p* = 0.030).

### 3.6. Associations of Physical Activity with GDM and FPG Values

We further examined the associations of both accelerometer-based and self-reported physical activity variables with the risk of GDM and the FPG value in 2nd trimester (see Table 4). While we did not find any associations of presence of GDM with accelerometer-based physical activity variables, multiple self-reported physical activity variables were found associated. Self-reported walking, total MPA, and total MVPA were, surprisingly, positively associated with presence of GDM (see Table 4 Model 1 for details) and, except for walking, the same held for other two physical activity variables after adjustment for demographic and other characteristics (Table 4 Model 2). These unexpected findings are probably due to less physical activity reported by those higher educated women, comprising 91% of the sample, who were at lower risk of GDM (see Appendix A and Table 3).

As to 2nd trimester FPG, it is found positively associated with accelerometer-measured light physical activity at 2nd trimester (Table 4 Model 1: e*^b^* = 1.001, 95% CI 1.0002 to 1.002, *p* = 0.023). The association attenuated after adjustment for demographic and other characteristics (Table 4 Model 2: e*^b^* = 1.001, 95% CI 1.00003 to 1.002, *p* = 0.044) and become no longer significant when additionally adjusting for 1st trimester FPG (Table 4 Model 3: e*^b^* = 1.001, 95% CI 0.9998 to 1.002, *p* = 0.089). It is notable that the positive effect of FPG in 1st trimester on that in 2nd trimester (*b* = 0.123, *p* = 0.030, Table 3) no longer existed when including accelerometer-measured light physical activity (*p* = 0.064, see Table 5). A similar pattern was observed for accelerometer-measured total physical activity (Table 4 and Table 5). No associations were found for accelerometer-measured MVPA nor any self-reported physical activity variables (Table 4).

To further investigate the above findings between physical activity variables and 2nd trimester FPG, we ad hoc analysed the role of physical activity during 2nd trimester in the association between 1st trimester FPG and 2nd trimester FPG. As shown in Table 3, a positive association of 1st trimester FPG with 2nd trimester FPG was found (*b* = 0.123, 95% CI 0.012 to 0.233, *p* = 0.030). We compared such association with the physical activity adjusted associations (Table 5) and found the positive effect of 1st trimester FPG reduced (*b* = 0.115 to 0.118) or disappeared (*p* > 0.050) when physical activity variables were included, except for accelerometer-based MVPA and self-reported non-walking MPA. For these two variables, the effect, surprisingly, became stronger (*b* = 0.128 for accelerometer-based MVPA) or held (*b* = 0.123 for self-reported non-walking MPA) (Table 5).

## 4. Discussion

This study quantified physical activity during pregnancy using both accelerometers and IPAQ and examined the relationship between these two measures. We found different patterns between physical activity measured by accelerometers and that measured by IPAQ and determined a non-linear relationship between these two measures. In addition, we further examined how these two types of physical activity measures were related to the risk of GDM as well as the change in FPG levels between 1st and 2nd trimesters, accounting for pregnant women’s demographic and pregnancy-related characteristics. Our findings suggested that physical activity of lower intensity may explain the different FPG values, and that physical activity engagement can potentially regulate FPG during early-to-mid pregnancy.

While the overall physical activity amount did not statistically reduce across pregnancy, accelerometer-based MVPA dropped from 1st trimester to 3rd trimester. This reflects the norm that high intensity of physical activity was gradually reduced in the course of pregnancy [6]. However, it is noteworthy that the sampled women were likely to be generally physically active as high levels of physical activity remained across pregnancy [15]. This may have contributed to the low GDM diagnosis rate since they had obtained benefits from physical activity prior to OGTT. The reason why the sample retained physically active could be due to clinicians’ in-pregnancy advice that staying physically active promotes better pregnancy outcomes [26]. The same reason may also explain why pregnant women more likely walked in 3rd trimester. While maintaining overall high PA amount, they shifted part of their non-walking MPA or intense physical activity to walking as an approach to remaining physically active.

We used both accelerometers and IPAQ to capture pregnant women’s physical activity, however, the values from these two measures considerably differ. By regressing accelerometer-based total physical activity (objectively measured) on self-reported total MVPA (subjective based on personal perception), we identified that pregnant women engaged more accelerometer-based physical activity than what they reported at lower levels of physical activity (<~1800 min per week) and tended to over-report their physical activity when at higher levels (>~1800 min per week). The association between objective and subjective measures may not be linear (Figure 3). These pieces of information are useful for future establishment of accelerometer-based physical activity intensity cut-points specific for pregnant women who are expected to demonstrate different patterns across trimesters which is distinct from those of non-pregnant adults. This can be further supported by comparison with the relationship between accelerometer-based MVPA and self-reported MVPA which yields poorer agreement (Appendix A).

Previous studies suggested mixed findings that physical activity during pregnancy can reduce GDM [11,27]. While some studies observed beneficial effects of physical activity, others found null results [11,27,28]. While the majority of previous studies were intervention studies, our study focused on free-living physical activity shed light on its potential benefits under real-life conditions of pregnant women. Given the variety of demographic and other characteristics among pregnant women, the beneficial effects of physical activity may be masked, and sometimes unexpected if demographic and other pregnancy-related characteristics are not considered. In our analyses, higher education was the major contributors to lower risk of GDM perhaps via other pathways prior to pregnancy rather than walking during pregnancy [29,30]. Increased MPA may not be essential for lowering GDM risk, particularly among women already physically active. However, we should be cautioned that these findings were derived from self-reports which may undergo subjective bias since accelerometer-based measures did not yield such findings. Towards a comprehensive understanding of physical activity in pregnancy, more studies on medium-high intensities of physical activity, especially in the second and third trimesters, are still encouraged.

Capitalising on the repeated measures of FPG in 1st and 2nd trimesters, we further examined the role of physical activity on the link between FPG at 1st trimester and that at 2nd trimester. Those women with higher 1st trimester FPG tended to engage in more physical activity in 2nd trimester, in particular those types of lower intensity, for example walking and light physical activity (see Appendix A), which could counteract the upward trend of FPG values from 1st to 2nd trimester. By comparing the relationships across the intensities of physical activity, light physical activity and walking were two main contributors, respectively, to objectively-measured and self-reported total physical activity since they yielded more consistent results with their corresponding total physical activity measures compared with other physical activity variables. This may imply that physical activity of lower intensity is more commonly performed during pregnancy and thus more suitable for pregnant women to accrue glucose-related health benefits. Future behavioural interventions can be targeted at lower intensity PA for pregnant women.

The consistent results from lower intensity physical activity between objective measure and self-report in relation to the changes in FPG values supports our postulation that part of present accelerometer-based light physical activity duration/time would more appropriately be classified into MPA as walking was classified as MPA in IPAQ (Figure 4). In addition, part of present accelerometer-based MPA is deemed vigorous to pregnant women, particularly in late pregnancy. In addition to the foresaid non-linear relationship between accelerometer-based and IPAQ-based total physical activity, this information should also be considered in identifying pregnant women-specific cut-points for accelerometer/device-measured physical activity.

Our study could not provide supporting evidence that physical activity can benefit FPG in a short-term fashion. However, based on a series of analyses in consideration of demographics and other covariates, we found particular characteristics (e.g., pre-pregnancy BMI) appear to be long-term determinants on pregnant women’s glucose level. Those with higher pre-pregnancy body weight appeared to engage in more physical activity during pregnancy in our sample, which may indicate their potential adherence to physical activity intervention. Demographic- or weight-specific physical activity interventions [27,31] for blood glucose management can also be considered in future.

This study has several strengths. Both accelerometer-based and self-reported physical activity were measured in a single sample to allow detailed comparison and the dose-response relationship between these two types of physical activity measures was able to be investigated. physical activity across three trimesters were captured. Covariate-adjusted and physical activity-adjusted changes in FPG values between 1st and 2nd trimesters were assessed to obtain more robust results. Limitations include that the causal relationships of physical activity variables with GDM and FPG cannot be inferred given the nature of the study design. Other factors related to the effect of physical activity on glucose intolerance or blood glucose level, such as diet, were not included. However, the measurement of dietary exposures has long been challenging.

## 5. Conclusions

In conclusion, different patterns in physical activity during pregnancy between objective measures and self-reports may help explain the mixed findings on physical activity –glucose relationships in literature. The identified non-linear relationships between objectively-measured total physical activity and self-reported total physical activity may inform establishment of accelerometer/device-based physical activity intensity classification specific to pregnant women and in turn improve the accuracy of physical activity measurement to yield unbiased assessment of such as physical activity intervention compliance. Given these advantages, using a set of wearable devices for capturing other aspects of physical activity or physical activity related behaviours could be considered in future studies. Our findings suggest that physical activity of lower intensity (light-to-moderate physical activity and walking) could be more appropriate for pregnant women to accrue glucose-related health benefits and that short-term increase in MPA prior to OGTT may not be necessary for reducing presence of GDM. Glucose management for pregnant women in future can be targeted at lower intensity physical activity.

## Figures and Tables

**Figure 1 ijerph-19-08064-f001:**
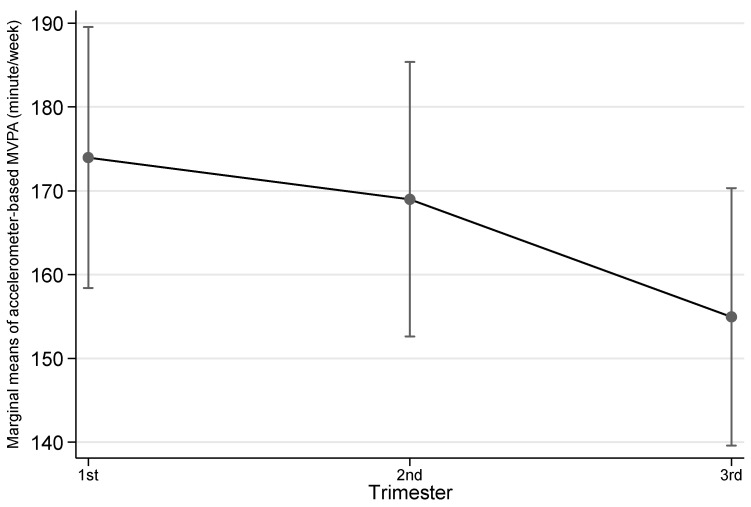
Accelerometer-based moderate-to-vigorous physical activity (MVPA) by trimester. Notes: The error bars represent 95% confidence intervals of the corresponding point estimates.

**Figure 2 ijerph-19-08064-f002:**
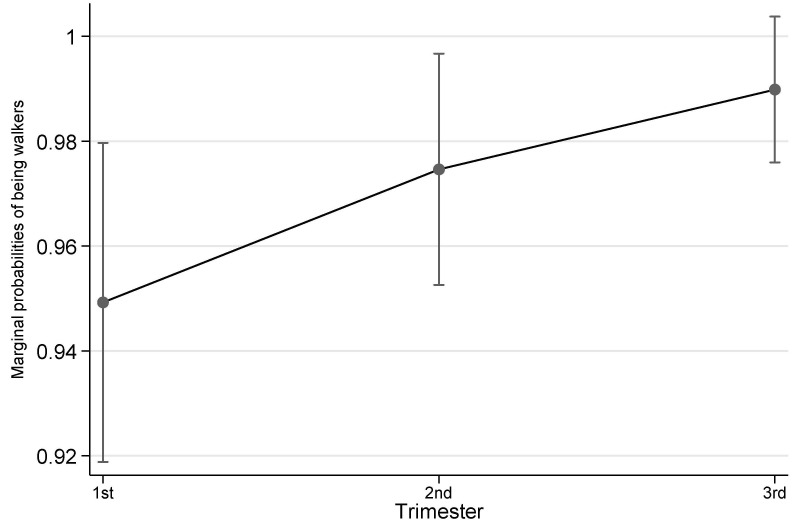
Probability of being walkers by trimester. Notes: The error bars represent 95% confidence intervals of the corresponding point estimates.

**Figure 3 ijerph-19-08064-f003:**
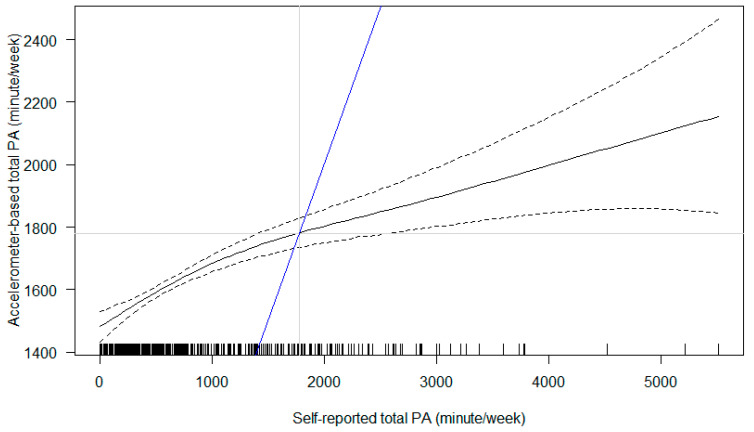
Association between self-reported total physical activity (PA) and accelerometer-based total PA. Notes: The black solid line represents point estimates (and the black dashed lines their 95% confidence intervals). The blue line is a reference line to indicate the hypothesised absolute agreement between two measures.

**Figure 4 ijerph-19-08064-f004:**
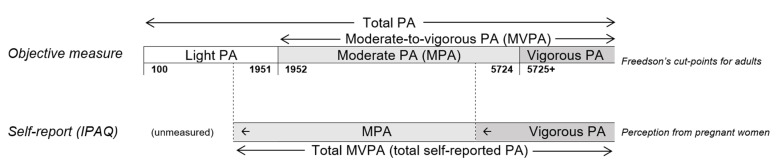
Hypothesised relationship between self-reported PA and accelerometer-based PA based on the scale of Freedson’s cut points. Notes: PA = physical activity. IPAQ = International Physical Activity Questionnaire. The positions of the cut-point only indicates the relative position but not their proportional positions.

**Table 1 ijerph-19-08064-t001:** Sample characteristics of pregnant women (*N* = 197).

Variable	*n* (%) or *M* (*SD*)	Range
Age, year	29.9 (3.07)	22–39
*Gestational age, week^+day^*		
1st trimester	12.6 (0.90)	10^+1^–14^+6^
2nd trimester	21.7 (1.04)	20^+0^–24^+5^
3rd trimester	31.0 (0.75)	30^+0^–33^+5^
*Educational attainment*		
Lower (up to secondary school)	18 (9.1%)	-
Higher (degree and above)	179 (90.9%)	-
*Household monthly income*		
Lower (less than 10,000 CNY)	102 (51.8%)	-
Higher (10,000 CNY and above)	95 (48.2%)	-
*Employment status*		
Full-time	159 (80.7%)	-
Other	38 (19.3%)	-
*Primipara*		
First	139 (70.6%)	-
Non-first	58 (29.4%)	-
*In vitro fertilisation*		
Yes	29 (14.7%)	-
No	168 (85.3%)	-
*Miscarriage history*		
Yes	38 (19.3%)	-
No	159 (80.7%)	-
*Pre-pregnancy BMI, kg/m^2^*	20.9 (3.05)	15.5–33.1
Underweight (<18.5 kg/m^2^)	42 (21.3%)	-
Normal (≥18.5 and <25 kg/m^2^)	129 (70.6%)	-
Overweight/obese (≥25 kg/m^2^)	16 (8.1%)	-
*Gestational diabetes mellitus*		
Yes	19 (9.6%)	-
No	178 (90.4%)	-
*Fasting plasma glucose level, mmol/L*		
1st trimester	3.99 (0.41)	3.0–5.7
2nd trimester (OGTT fasting)	4.20 (0.32)	3.5–5.6

CNY = Chinese yuan; BMI = body mass index; OGTT = oral glucose tolerance test. ‘-’ indicates not applicable.

**Table 2 ijerph-19-08064-t002:** Physical activity (PA) measures during pregnancy (*N* =197, with 591 observations across three trimesters).

Variables (Unit: Minutes/Week)	*M* (*SD*)	Medium (IQR) ^a^	Range	For Those Engaging in PA ^(i)^ *p* Value	None vs Any PA Engagement ^(ii)^ *p* Value
*Accelerometer-based ^(iii)^*					
**Light PA**					
1st trimester	1469 (429.5)	-	546–2875	Reference	-
2nd trimester	1481 (416.7)	-	651–2806	0.196	-
3rd trimester	1463 (395.4)	-	666–2972	0.290	-
**MVPA**					
1st trimester	177 (113.5)	167 (137) ^b^	2–604	Reference	-
2nd trimester	168 (116.4)	148 (118)	1–806	0.430	-
3rd trimester	153 (110.5)	130 (147)	1–584	0.009	-
**Total PA**					
1st trimester	1646 (467.3)	-	551–3031	Reference	-
2nd trimester	1649 (436.5)	-	695–2963	0.331	-
3rd trimester	1616 (426.3)	-	670–3154	0.802	-
*Self-reported*					
**Walking**					
1st trimester	660 (752.0)	410 (660)	0–4620	Reference	Reference
2nd trimester	717 (720.4)	450 (820)	0–4410	0.466	0.176
3rd trimester	633 (648.3)	420 (660)	0–3780	0.284	0.027
**Non-walking MPA**					
1st trimester	148 (243.0)	30 (200)	0–1200	Reference	Reference
2nd trimester	215 (361.6)	70 (240)	0–2050	0.085	0.062
3rd trimester	173 (285.9)	60 (200)	0–1600	0.456	0.541
**Total MPA (walking plus non-walking MPA)**					
1st trimester	808 (824.7)	500 (955)	0–5220	Reference	-
2nd trimester	931 (860.4)	610 (1070)	0–4590	0.146	-
3rd trimester	805 (726.7)	570 (870)	0–3790	0.971	-
**Total MVPA**					
1st trimester	818 (835.0)	500 (955)	0–5520	Reference	-
2nd trimester	951 (888.0)	610 (1090)	0–5220	0.127	-
3rd trimester	811 (730.3)	570 (860)	0–3790	0.923	-

Notes: *M* = mean; *SD* = standard deviation; IQR = interquartile range; PA = physical activity; MPA = moderate physical activity; MVPA = moderate-to-vigorous physical activity. ‘-’ indicates not applicable. ^a^ computed for those with skewness >|1.0|; ^b^ not skewed but computed for the purpose of comparison with other trimester subgroups. ^(i)^ Based on generalised linear models with gamma variance and logarithmic link function (for accelerometer-based), negative binominal models (for self-reported, except walking and non-walking MPA), and zero-inflated negative binominal models (for walking and non-walking MPA); ^(ii)^ Only applicable to zero-inflated negative binominal models; ^(iii)^ Adjusted for wear time.

**Table 3 ijerph-19-08064-t003:** Adjusted (multivariable) associations of sociodemographic and health-related characteristics with gestational diabetes mellitus (GDM) and other glucose measures.

	GDM ^(i)^ (*N* = 197)			1st Trimester FPG ^(ii)^ (*N* = 197)			2nd Trimester FPG (OGTT Fasting) ^(ii)^ (*N* = 196)		
Variable (Unit)	a*OR*	95% CI	*p*	e*^b^*	95% CI	*p*	e*^b^*	95% CI	*p*
*Gestational age (week)*	0.975	(0.598, 1.590)	0.919	0.975	(0.916, 1.037)	0.419	1.017	(0.975, 1.060)	0.440
*Age (year)*	1.075	(0.890, 1.297)	0.454	0.996	(0.975, 1.018)	0.751	0.9999	(0.983, 1.017)	0.989
*Educational attainment*									
Lower	Reference			Reference			Reference		
Higher	0.211	(0.047, 0.954) *	0.043	1.022	(0.829, 1.260)	0.840	0.990	(0.842, 1.165)	0.907
*Household monthly income*									
Lower	Reference			Reference			Reference		
Higher	1.152	(0.397, 3.343)	0.795	1.106	(0.984, 1.242)	0.091	1.056	(0.963, 1.157)	0.242
*Employment status*									
Full-time	2.970	(0.471, 18.742)	0.247	1.104	(0.943, 1.292)	0.220	0.981	(0.867, 1.109)	0.756
Other	Reference			Reference			Reference		
*Primipara*									
Non-first	Reference			Reference			Reference		
First	3.627	(0.852, 15.443)	0.081	1.008	(0.874, 1.163)	0.909	1.042	(0.933, 1.164)	0.464
*In vitro fertilisation*									
No	Reference			Reference			Reference		
Yes	0.220	(0.0245, 1.978)	0.177	0.9996	(0.836, 1.195)	0.996	0.985	(0.858, 1.132)	0.836
*Miscarriage history*									
No	Reference			Reference			Reference		
Yes	1.987	(0.604, 6.533)	0.258	1.030	(0.890, 1.193)	0.690	1.024	(0.914, 1.148)	0.682
*Pre-pregnancy BMI (kg/m^2^)*	1.124	(0.966, 1.309)	0.131	1.025	(1.006, 1.045) *	0.011	1.032	(1.016, 1.047) ***	<0.001
Normal	Reference ^#^			Reference ^#^			Reference ^#^		
Underweight	0.176	(0.022, 1.407)	0.101	0.862	(0.747, 0.996) *	0.043	0.868	(0.774, 0.973) *	0.015
Overweight/Obese	0.426	(0.0467, 3.886)	0.449	1.117	(0.903, 1.382)	0.307	1.180	(0.998, 1.395)	0.053
							* **b** *	**95% CI**	* **p** *
*1st trimester FPG*	-	-	*-*	-	-	*-*	0.123	(0.012, 0.233) *	0.030

Notes: a*OR* = adjusted odds ratio; CI = confidence interval; *p* = *p* value; BMI = body mass index; FPG = fasting plasma glucose; e*^b^* = antilogarithm of regression coefficient, interpreted as the proportional increase (if >1) or decrease (if <1) in outcome variable associated with a 1-unit increase in predictor variable. ‘-’ indicates not applicable. ^(i)^ Estimates are obtained from logistic regression. ^(ii)^ Estimates are obtained from general linear model. ^#^ When this categorised variable was included in the model, the estimates of other variables slightly differ. Nonetheless, the estimates reported in this table are those with the original (uncategorised) BMI. * *p* < 0.05, *** *p* < 0.001. To avoid confusion, more than three decimal digits are reported (as 1.000 would have been shown if the estimate was rounded to three decimal digits).

**Table 4 ijerph-19-08064-t004:** Association of physical activity (PA) variables at 2nd trimester with diagnosis of gestational diabetes mellitus (GDM) and 2nd trimester fasting plasma glucose (FPG) (*N* = 197).

	GDM ^(i)^						2nd Trimester FPG ^(ii)^								
	Model 1			Model 2			Model 1			Model 2			Model 3		
PA Variable (Unit: 10 Minutes/Week)	a*OR*	95% CI	*p*	a*OR*	95% CI	*p*	e*^b^*	95% CI	*p*	e*^b^*	95% CI	*p*	e*^b^*	95% CI	*p*
*Accelerometer-based* ^(iii)^															
Light PA	1.008	(0.997, 1.019)	0.174	1.009	(0.996, 1.021)	0.179	1.001	(1.0002, 1.002) *	0.023	1.001	(1.00003, 1.002) *	0.044	1.001	(0.9998, 1.002)	0.089
MVPA	1.014	(0.998, 1.005)	0.480	1.001	(0.996, 1.005)	0.745	1.002	(0.998, 1.006)	0.412	1.001	(0.997, 1.006)	0.484	1.002	(0.998, 1.006)	0.347
Total PA	1.008	(0.997, 1.019)	0.136	1.008	(0.996, 1.021)	0.171	1.001	(1.0003, 1.002) *	0.015	1.001	(1.0001, 1.002) *	0.032	1.001	(0.99997, 1.002)	0.058
*Self-reported* ^(vi)^	*OR*	95% CI	*p*												
Walking	1.006	(1.001, 1.011) *	0.029	1.005	(0.998, 1.011)	0.148	1.0002	(0.9996, 1.001)	0.452	1.0003	(0.9996, 1.001)	0.401	1.0002	(0.9995, 1.001)	0.580
Non-walking MPA [*n* = 134]	1.009	(0.998, 1.019)	0.100	1.014	(0.9998, 1.028)	0.054	1.001	(0.9999, 1.002)	0.073	1.001	(0.9998, 1.002)	0.085	1.001	(0.99987, 1.002)	0.079
None [*n* = 63]	Reference			Reference			Reference			Reference			Reference		
Any non-walking MPA [*n* = 134]	2.712	(0.760, 9.673)	0.124	3.386	(0.813, 14.094)	0.094	0.994	(0.903, 1.095)	0.908	0.960	(0.870, 1.058)	0.411	0.959	(0.871, 1.057)	0.401
Total MPA	1.006	(1.002, 1.010) **	0.006	1.006	(1.001, 1.012) *	0.024	1.0004	(0.9998, 1.001)	0.167	1.0004	(0.9999, 1.0009)	0.154	1.0003	(0.9998, 1.0009)	0.231
Total MVPA	1.006	(1.002, 1.010) **	0.006	1.006	(1.001, 1.012) *	0.021	1.0003	(0.9998, 1.0001)	0.184	1.0004	(0.9998, 1.001)	0.179	1.0003	(0.9998, 1.001)	0.259

Notes: *OR* = odds ratios; a*OR* = adjusted odds ratio; e*^b^* = antilogarithm of regression coefficient, interpreted as the proportional increase (if >1) or decrease (if <1) in outcome variable associated with a 1-unitincrease in independent variable; CI = confidence interval; *p* = *p* value; BMI = body mass index. Model 1: Only adjusted for wear time (accelerometer-based only). Model 2: Model 1 + covariates. Model 3: Model 2 + 1st trimester FPG. ^(i)^ Estimates are obtained from logistic regression. ^(ii)^ Estimates are obtained from general linear model. ^(iii)^ Covariates include wear time, gestational week, age, education attainment, household monthly income, employment status, primipara, in vitro fertilisation, miscarriage history, pre-pregnancy BMI. ^(vi)^ Covariates include gestational week, age, education attainment, household monthly income, employment status, primipara, in vitro fertilisation, miscarriage history, pre-pregnancy BMI. * *p* < 0.050, ** *p* < 0.010. To avoid confusion, more than three decimal digits are reported (as 1.000 would have been shown if the estimate was rounded to three decimal digits).

**Table 5 ijerph-19-08064-t005:** Physical activity (PA) adjusted association between 1st trimester FPG and 2nd trimester FPG (*N* = 196).

Outcome Variable: 2nd Trimester FPG																					
	Accelerometer-Based									Self-Reported											
	Light PA			MVPA			Total PA			Walking			Non-Walking MPA			Total MPA			Total MVPA		
	*b*	95% CI	*p*	*b*	95% CI	*p*	*b*	95% CI	*p*	*b*	95% CI	*p*	*b*	95% CI	*p*	*b*	95% CI	*p*	*b*	95% CI	*p*
1st trimester FPG	0.106	(−0.006, 0.219)	0.064	0.128	(0.016, 0.240) *	0.025	0.108	(−0.003, 0.220)	0.057	0.118	(0.006, 0.230) *	0.039	0.123	(0.013, 0.234) *	0.028	0.115	(0.003, 0.226) *	0.043	0.116	(0.004, 0.227) *	0.042

Notes: *b* = regression coefficient; CI = confidence interval; *p* = *p* value. All estimates are adjusted for wear time (accelerometer-based), gestational week, age, education attainment, household monthly income, employment status, primipara, in vitro fertilisation, miscarriage history, and pre-pregnancy BMI. * *p* < 0.050.

## Data Availability

The data presented in this study are available on request from the corresponding author. The data are not publicly available due to institutional data protection policies.

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
