# Peer review of "Physical Activity during Pregnancy: Comparisons between Objective Measures and Self-Reports in Relation to Blood Glucose Levels"

_ijerph, 2022, doi:10.3390/ijerph19138064_

Round 1

Reviewer 1 Report

Recommendations:
There is an excessive use of acronyms both in the abstract and in the text in general which makes it very difficult to read.
In the introduction, it is recommended that the results of previous research relating physical activity and diabetes be included to justify the importance of their measurement.
The authorisation of the ethics committee of the body in which the study is carried out is missing.
It is recommended that the statistical package used to process the data be specified and that the analysis of the variables be better structured for a better understanding. 

Reviewer 3 Report

The manuscript is interesting because it emphasizes the importance of practicing physical activity during pregnancy. In my opinion, an accelerometer is a tool that allows monitoring in "real life", therefore more reliable than questionnaires alone. Still, the positioning on the wrist and the only measurement of acceleration parameters (and not heartbeat, body temperature, impedance) make the work possibly affected by errors, it could be considered to use a more complete instrument, in order to have an estimate of consumption energy, the intensity of physical exercise and also the quality of sleep.

This should be emphasized in the conclusions.

For the rest, the work is interesting and could be the first step towards a more careful analysis of the physical activity in pregnancy, which should be proposed, where possible, even at medium-high intensities, especially in the second and third trimesters.
